# Proline Protects Boar Sperm against Oxidative Stress through Proline Dehydrogenase-Mediated Metabolism and the Amine Structure of Pyrrolidine

**DOI:** 10.3390/ani10091549

**Published:** 2020-09-01

**Authors:** Chengwen Feng, Zhendong Zhu, Wenjing Bai, Rongnan Li, Yi Zheng, Xiu’e Tian, De Wu, Hongzhao Lu, Yongjun Wang, Wenxian Zeng

**Affiliations:** 1Key Laboratory for Animal Genetics, Breeding and Reproduction of Shaanxi Province, College of Animal Science and Technology, Northwest A&F University, Yangling 712100, Shaanxi, China; fengchengwen585@163.com (C.F.); zhuzhengdong2014@163.com (Z.Z.); baiwenjing999@163.com (W.B.); lirongnan2016@nwafu.edu.cn (R.L.); y.zheng@nwafu.edu.cn (Y.Z.); txe82@nwafu.edu.cn (X.T.); 2Key Laboratory for Animal Disease Resistance Nutrition of the Ministry of Education of China, Institute of Animal Nutrition, Sichuan Agricultural University, Wenjiang, Chengdu 610000, Sichuan, China; wude@sicau.edu.cn; 3College of Biological Science and Engineering, Shaanxi University of Technology, Hanzhong 723001, Shaanxi, China; zl780823@126.com

**Keywords:** proline, ROS, secondary amine, PRODH, boar semen

## Abstract

**Simple Summary:**

Reactive oxygen species that accumulate during liquid storage of boar semen lead to oxidative stress to sperm. In this study, we found that proline significantly improved boar sperm quality and protected sperm against oxidative damages during liquid storage at 17 °C. Using the model of artificially induced oxidative stress, we found that proline exerted an antioxidative role by modulating redox homeostasis in boar sperm. The secondary amine structure of proline and proline dehydrogenase-mediated metabolism are involved in the antioxidative role. We suggest that addition of proline to the extender would be beneficial to improve boar sperm quality.

**Abstract:**

Proline was reported to improve sperm quality in rams, stallions, cynomolgus monkeys, donkeys, and canines during cryopreservation. However, the underlying mechanism remains unclear. The aim of this study was to investigate the effect of proline on boar semen during liquid storage at 17 °C and explore the underlying mechanism. Freshly ejaculated boar semen was supplemented with different concentrations of proline (0, 25, 50, 75, 100, 125 mM) and stored at 17 °C for nine days. Sperm motility patterns, membrane integrity, ATP (adenosine triphosphate), reactive oxygen species (ROS), and GSH (glutathione) levels, and the activities of catalase (CAT) and superoxide dismutase (SOD) were evaluated after storage for up to five days. It was observed that boar sperm quality gradually decreased with the extension of storage time, while the ROS levels increased. Addition of 75 mM proline not only significantly improved sperm membrane integrity, motility, and ATP levels but also maintained the redox homeostasis via increasing the GSH levels and activities of CAT and SOD. When hydrogen peroxide (H_2_O_2_) was used to induce oxidative stress, addition of proline significantly improved sperm quality and reduced ROS levels. Moreover, addition of proline also improved sperm quality during the rapid cooling process. Notably, addition of DL-PCA (DL-pipecolinic acid) rescued the reduction of progressive motility and total motility caused by H_2_O_2_, and THFA (tetrahydro-2-furoic acid) failed to provide protection. Furthermore, addition of proline at 75 mM increased the activity of proline dehydrogenase (PRODH) and attenuated the H_2_O_2_-induced reduction in progressive motility. These data demonstrate that proline protects sperm against oxidative stress through the secondary amine structure and proline dehydrogenase-mediated metabolism.

## 1. Introduction

Artificial insemination is widely used in pig industry for improving the fertility and spreading genetic progress [1,2]. In the farm and boar station, the semen is diluted and stored at 17 °C in vitro [3,4,5]. However, the sperm fertility decreases with the extension of preservation [6,7]. One of the major issues is the accumulation of reactive oxygen species (ROS) [6,8,9,10]. The uncontrolled and excessive production of ROS that overwhelms the limited antioxidant defense system in sperm and seminal plasma leads to oxidative stress to sperm [10,11].

Unfortunately, the plasma membrane of sperm is rich in polyunsaturated fatty acids, which results in the membranes being susceptible to oxidative stress [7,10,12]. Meanwhile, the antioxidant defense system in sperm is very limited [13,14,15]. Therefore, sperm is prone to lipid peroxidation (LPO) [6,16] and DNA damage [16,17] under oxidative stress. Moreover, excessive ROS damages the membrane structure and causes irreversible loss of motility [18,19,20]. Thus, maintaining the balance of ROS generation and scavenging is essential to maintain cell structure and function. The addition of antioxidants protects sperm from oxidative damages and extends the lifespan [21,22].

Proline has been reported to protect plant cells, *Saccharomyces cerevisiae*, and mammalian cells against oxidative stress [23,24,25]. The accumulation of proline seems to be an adaptive stress response in responding to ROS stress. The concentration of proline in plant cells increases up to 100 times when they are exposed to UV or heavy metal stress [23,26,27]. Meanwhile, mammalian somatic cells enhance proline biosynthesis when exposed to hydrogen peroxide (H_2_O_2_) [24]. Interestingly, Natarajan et al. [28] reported that treatment of melanoma cells with exogenous proline significantly increased cell viability and diminished oxidative damage under ROS stress.

Growing evidence has demonstrated that ROS accumulates during the cooling and freezing-thawing process [29]. Addition of proline to semen extender significantly improved donkey sperm motility after cold storage at 5 °C for three days [30]. Supplementation of proline to the freezing extender also improved motility [31,32,33,34,35,36] and decreased lipid peroxidation damage in ram sperm [31] during the cryopreservation process. Similarly, addition of 5 mM proline significantly improved motility, membrane, and acrosome integrity after thawing in cynomolgus monkey sperm [37]. In addition, proline protected sperm from cryoinjury in donkeys [38] and canines [39]. However, the underling mechanisms of the protective role of proline on sperm are largely unknown. Therefore, the aim of the present study was to investigate whether the addition of the proline could improve boar sperm quality, and, if so, to elucidate the underling mechanism.

## 2. Materials and Methods

### 2.1. Reagents and Media

Unless otherwise stated, all chemicals were purchased from Sigma-Aldrich (Shanghai, China).

### 2.2. Ethical Approval

The procedures for care and use of animals were approved by the Institutional Animal Care and Use Committee of Northwest A&F University (ethical approval code: H17-09).

### 2.3. Animals and Semen Collection

Five healthy and fertile Duroc boars aged 1.5 to 2 years were used in this study. The boars were housed individually, fed with basal diets, and allowed free access to water. Boar semen was acquired weekly by filtrating through double gauze after collecting the sperm-rich fraction of ejaculate via the gloved-hand technique. Ejaculated semen was collected into flasks at 37 °C and processed in the laboratory within 30 min after collection. Only ejaculates containing more than 90% motile sperm (evaluated by contrast-phase microscopy) were used in this study. To avoid individual differences, the ejaculated semen was pooled as an independent sample. Analyses (*n* = 3) were performed with three independent samples collected over three consecutive weeks.

### 2.4. Experiment I

Experiment I was designed to explore the effect of proline on boar sperm during liquid storage. As described below, semen was diluted with modified Modena extender containing different concentrations of proline (0, 25, 50, 75, 100, 125 mM). The extended boar semen was stored at 17 °C. Sperm motility was evaluated every other day. The membrane and acrosome integrity, ATP levels, and MMP (mitochondrial membrane potential) were evaluated at day 5 of the storage. To explore whether the addition of proline could maintain sperm lifespan, sperm motility was monitored at day 9 of the storage.

#### 2.4.1. Semen Storage

The mixed fresh semen was first isothermally diluted to 100 million/mL with Modena extender, which is composed of 152.8 mM D-glucose, 26.7 mM trisodium citrate, 15.1 mM citric acid, 6.3 mM EDTA disodium, 11.9 mM sodium hydrogen carbonate, 46.6 mM Tris, 400 IU/mL Polymyxin B (Amersco, Shanghai, China), 1000 IU/mL penicillin G sodium salt (Solarbio, Beijing, China), and 1 mg/mL streptomycin sulfate (Solarbio, Beijing, China; pH = 7.2). These samples were then divided into six equal fractions (7 mL per fraction), incubated with various concentrations of proline (TCI, Shanghai, China), respectively, and stored at 17 °C for up to 9 days (BC-43KT1, Hisense Co., Beijing, China).

#### 2.4.2. Motility Analysis

Sperm motility was evaluated via a computer-assisted sperm analysis (CASA) system (HVIEW, Fuzhou, China) every two days from the first day of storage. Briefly, samples were transferred into a water bath at 37 °C for 10 min, then 6 μL semen were dropped onto a pre-heated glass slide (37 °C; 20 μm CELL-VU^®^ DRM-600 sperm count slide; Millennium Sciences, NewYork, NY, USA) for evaluation according to Zhu et al. [20]. The standard parameter settings in this experiment were 30 frames/s. The percentage of sperm with VCL (curvilinear velocity) of >10 μm/s was described as the total motility and percentage of sperm that exhibited a VSL (straight line velocity) of >25 µm/s and an STR (straightness) of ≥75% was described as progressive motility. At least five fields were evaluated for each sample and every treatment was repeated independently in triplicates. As for each sample, at least 1000 sperm were evaluated for motility.

#### 2.4.3. Integrity of Membrane and Acrosome

According to Zhu et al. [29,40], the LIVE/DEAD™ Sperm Viability Kit (Invitrogen™, Shanghai, China) and fluoresce isothiocyanate-peanut agglutinin (FITC-PNA) were used for the evaluation of membrane integrity and acrosome integrity, respectively. Sperm samples were incubated with SYBR working solution (100 nM) at 37 °C for 5 min and then with PI (propidium iodide) working solution (12 μM) for another 5 min for the detection of membrane integrity. As for the detection of acrosomal integrity, samples were smeared on a clean glass slide, air-dried, and fixed in absolute methanol for 10 min. Subsequently, samples were stained with fluoresce isothiocyanate-peanut agglutinin (FITC-PNA; 100 μg/mL in PBS, namely phosphate buffer solution) in moist and dark conditions at 37 °C for 30 min. Stained samples were observed and photographed with a fluorescence microscope (Nikon 80i, Tokyo, Japan) at 400× magnification after staining. Sperm stained with PI were photographed at 535 nm excitation and 617 nm emission, while sperm stained with SYBR-14 or FITC-PNA were photographed at 488 nm excitation and 525 nm emission. A minimum of 200 sperm in each field and at least 5 random fields were evaluated for each sample.

#### 2.4.4. Measurement of ATP Levels

Sperm ATP levels were evaluated with the ATP Assay Kit (Beyotime Institute of Biotechnology, Shanghai, China) following the manufacturer’s instruction. Briefly, sperm samples (10^7^ sperm) were lysed with 200 μL of schizolysis solution as well as ultrasonication (20 KHz, 750 W, operating at 40% power, five cycles for 3 s on and 5 s off), and then centrifuged at 4 °C for 10 min at 12,000× *g*. Supernatant was retained for subsequent detection. Then, 100 µL of working solution for ATP detection was added to a 96-well plate, placed at room temperature for 5 min, and mixed with 20 μL of supernatant. The luminescence at integration × 1000 ms was read by an ascent luminoskan luminometer (Thermo Scientific, Palm Beach, FL, USA) with BPSE (Bio-Protein-Specific Enzyme) as a blank for each experiment.

#### 2.4.5. Mitochondrial Membrane Potentials

A Mitochondrial Membrane Potential (MMP) Detection Kit (Beyotime Institute of Biotechnology, Shanghai, China) was used to evaluate the sperm mitochondrial membrane potential. JC-1 (5, 5′, 6, 6′-tetrachloro-1, 1′, 3, 3′-tetraethylbenzimidazolcarbocyanine iodide) aggregates in the matrix of mitochondria and shows red fluorescence when MMP is high. By contrast, it exists in monomer form and shows green fluorescence when MMP is low. Briefly, sperm samples were stained with JC-1 solution (28 µL of stock solution diluted in 72 µL of Modena solution) at 37 °C for 30 min in the dark. Samples were then centrifuged at 600× *g* for 5 min and re-suspended with JC-1 buffer on ice. The stained spermatozoa were detected at 490 nm excitation and 530 nm emission for green fluorescence while 525 nm excitation and 590 nm emission for red fluorescence via a multimode reader (Synergy HT, Bio Tek, Winooski, VT, USA). The ratio of the fluorescence intensity of sperm with high MMP to those with low MMP is shown to reflect the activity of mitochondria.

### 2.5. Experiment II

Experiment II was designed to explore whether proline modulated redox homeostasis in boar sperm. Sperm samples were treated following the protocol described in the “semen incubation” subsubsection in experiment I and stored at 17 °C for 5 days. ROS levels, LPO levels, GSH (glutathione) levels, and the activity of CAT and SOD were evaluated at the fifth day of storage after treatment.

#### 2.5.1. Reactive Oxygen Species

The Reactive Oxygen Species Assay Kit (S0033S; Beyotime Institute of Biotechnology, Shanghai, China) was used for the evaluation of the total ROS level following the manufacturer’s instruction. Briefly, sperm samples were washed three times with Modena solution and re-suspended with 10 µM 2′,7′-Dichlorodihydrofluorescein diacetate (DCFH-DA) working solution, then incubated at 37 °C for 30 min in the dark. Fluorescence intensity was detected by a multimode reader (Synergy HT, Bio Tek) at 488 nm excitation and 525 nm emission.

#### 2.5.2. Lipid Peroxidation

BODIPY 581/591C11 (Invitrogen™, Shanghai, China) was used for the assessment of lipid peroxidation following the protocol described by Zhu et al. [41]. Briefly, 200 µL of each sample were washed twice with Modena, re-suspended with BODIPY 581/591C11 working solution (10 μM), and incubated at 37 °C for 30 min in the dark. Subsequently, samples were washed twice with Modena solution to remove those unbound dyes. The relative fluorescence intensity of the samples was quantified via a multimode reader (Synergy HT, Bio Tek) at 488 nm excitation/525 nm emission (oxidized) and 595 nm excitation/625 nm emission (unoxidized), respectively. The ratio of the fluorescence intensity of oxidized dye to unoxidized one reflects the degree of lipid peroxidation.

#### 2.5.3. Measurement of Glutathione Levels

Total glutathione (GSH) levels were evaluated following the protocol described by Zhu et al. [40] through a total glutathione assay kit (S0052, Beyotime Institute of Biotechnology, Shanghai, China). Briefly, sperm samples were centrifuged at 1000× *g* for 5 min at room temperature, re-suspended with a three-fold volume of Modena solution, and centrifuged at 1000× *g* for 10 min after cytolysis by three cycles of rapid cooling in liquid nitrogen and thawing at 37.8 °C. The supernatant was transferred into a 96-well plate to evaluate the absorption at 412 nm.

#### 2.5.4. Analysis of Catalase and Superoxide Dismutase Activity

A total superoxide dismutase assay kit with WST-8 (S0101, Beyotime Institute of Biotechnology, Shanghai, China) and catalase assay kit (S0051, Beyotime Institute of Biotechnology, Shanghai, China) were used to evaluate the activity of SOD and CAT, respectively. Briefly, sperm samples were washed three times with Modena solution and centrifuged at 12,000× *g* for 10 min at 4 °C after ultrasonication (20 KHz, 750 W, operating at 40%, five cycles for 5 s on and 5 s off) on ice. Then, supernatant was collected for the detection of catalase and superoxide dismutase activity following the manufacturer’s instruction.

### 2.6. Experiment III

Experiment III was designed to further explore whether proline improved boar sperm quality by quenching ROS. Sperm samples were treated with different concentrations of proline in the case of intentionally induced oxidative stress. The oxidative stress was directly induced via incubating sperm with 200 μM H_2_O_2_ (37 °C, 2 h) or via rapid cooling that would allow ROS accumulation, respectively. Sperm quality (membrane and acrosome integrity and MMP) and redox parameters (ROS levels, LPO levels, GSH levels, and activity of CAT and SOD) were measured according to the corresponding protocols mentioned above.

#### Rapid Cooling

The method for rapid cooling described by Zeng and Terada was used after a slight modification in this experiment [42]. Briefly, semen samples were washed twice with Modena solution by centrifugation at 800× *g* for 5 min. Sperm samples were then adjusted to the concentration of 50 million/mL with Modena solution containing different concentrations of proline. Subsequently, samples were incubated at 30 °C for 15 min and transferred into 5 °C water immediately for 15 min, while the samples in the Bc group (before cooling) were incubated at 30 °C all the time. Sperm samples were then treated differently according to the parameters to be detected.

### 2.7. Experiment IV

Experiment IV was designed to explore whether the secondary amine of pyrrolidine was the key factor for the protective role of proline against oxidative stress, and the proline analogs that contain the secondary amine or not were added to the extender. Among those analogs, DL-pipecolinic acid (DL-PCA) was selected as a positive substrate because it contains a secondary amine structure while tetrahydro-2-furoic acid (THFA) as a negative one. DL-pipecolinic acid (DL-PCA, Shanghai Yuanye Bio-technology Co. Ltd., Shanghai, China) or tetrahydro-2-furoic acid (THFA, TCI, Shanghai, China) was added to the extender before exposure to H_2_O_2_ (200 μM, 37 °C, 2 h). Sperm motility and progressive motility were detected according to the protocol mentioned in the “motility analysis” subsubsection.

### 2.8. Experiment V

Experiment V was designed to verify whether proline exerts an antioxidative role via proline dehydrogenase (PRODH), which is the enzyme that catalyzes the first step of proline catabolism. The expression of PRODH and its subcellular location were first identified via Western blotting and immunofluorescence staining. PRODH activity was detected at the fifth day of storage after treatment according to the followed protocol mentioned in the “analysis of proline dehydrogenase activity” subsubsection. Tetrahydro-2-furoic acid, a specific inhibitor of PRODH, was added to the extender to verify the role of PRODH in the protection role of proline.

#### 2.8.1. Western Blotting

Protein samples were prepared via ultrasonication (20 KHz, 750 W, operating at 30% power, six cycles for 5 s on and 5 s off) after the addition of proteinase inhibitors. The total protein (20 μg) of each sample was separated by 10% SDS-PAGE gel and transferred onto a polyvinylidene fluoride (PVDF) membrane. After incubation with 5% (*m*/*v*) non-fat dried milk in TBST (0.02 mM Tris, 0.14 mM NaCl, 0.1% Tween 20) for 2 h, these membranes were immunoblotted with primary antibodies against PRODH (22980-1-AP, 1:3000, Proteintech, Wuhan, China) or tubulin (11224-1-AP, 1:3000, Proteintech, Wuhan, China) separately at 4 °C for 12 h. Subsequently, membranes were washed with TBST solution, incubated with a secondary antibody (horseradish peroxidase conjugated goat anti-rabbit IgG, 1:3000; CWBIO, Beijing, China) for 2 h, and washed with TBST solution for another 30 min. The reagent for enhanced chemiluminescence (MA 01821, Millipore Corporation, Billerica, MA, USA) was used for detection and developed by an X-ray film.

#### 2.8.2. Immunofluorescence Staining of PRODH in Boar Sperm

Aliquots of 100 μL (1 × 10^7^ sperm per mL) sperm samples were fixed with 4% paraformaldehyde for 15 min, washed with PBS, and smeared on a clean slide. These samples were then permeabilized with 0.25% (*v*/*v*) Triton X-100 in PBS for 15 min and washed again with PBS. Subsequently, boar sperm was blocked with 10% donkey serum (*v*/*v*) at 37 °C for 2 h and incubated with a primary antibody against PRODH (22980-1-AP, 1:3000, Proteintech, Wuhan, China) at 4 °C overnight. These samples were washed and resuspended with a secondary antibody (FITC-conjugated goat anti-rabbit IgG, 1:200; CWBIO, Beijing, China) for 2 h at 4 °C. Finally, samples were washed with PBS, stained with DAPI (4,6-diamidino-2-phenylindole, 1:1000; Beyotime Institute of Biotechnology, Shanghai, China) for 10 min, and sealed with 40% glycerin. Fluorescent images were obtained with a fluorescence microscope (Nikon 80i, Tokyo, Japan).

#### 2.8.3. Analysis of Proline Dehydrogenase Activity

A proline dehydrogenase (PRODH) assay kit (BC4165, Beijing solarbio science & technology Co., Ltd., Beijing, China) was used to detect the activity of PRODH in boar sperm according to the manufacturer’s instruction. In brief, aliquots of 100 μL (1 × 10^8^ sperm per mL) sperm samples were washed twice with PBS by centrifugation at 3000× *g* for 3 min, lysed with 1 mL of extracting solution I and ultrasonication (20 KHz, 750 W, operating at 30% power, six cycles for 5 s on and 5 s off), and then centrifuged at 1500× *g* for 15 min at 4 °C. Subsequently, the supernatant was incubated with 10 μL of extracting solution II on ice for 30 min and centrifuged at 15,000× *g* for 20 min at 4 °C. The mixtures of working solution and the supernatant were transferred to a 96-well plate to evaluate the absorption at 600 nm at 10 s after blending and correspondingly evaluate the absorption at 600 nm at 190 s after an incubation of the mixture for 3 min at 37 °C.

### 2.9. Statistical Analysis

All parameters were evaluated at least three times with independent samples. Data were tested for normality and variance homogeneity using the Shapiro–Wilk and Levene’s tests, respectively, prior to statistical analysis. A *p*-value greater than the significance level of 0.05 means that the data follow a normal distribution. If necessary, the data were transformed with arc-sin square root transformation. Data were analyzed by a general mixed model (with repeated measures), followed by multiple comparisons with the Tukey test by using SPSS version 19.0 for Windows (SPSS Inc., Chicago, IL, USA). In this model, the treatment was the inter-subject factor, and storage time was the intrasubject factor. In all cases, each functional parameter was the dependent variable. Results are expressed as mean ± standard error of the mean (SEM). *p* < 0.05 was considered significantly different in the statistics.

## 3. Results

### 3.1. Experiment I

#### 3.1.1. Proline Improves Sperm Motility

Motility acquisition is essential for the function and ultimately male fertility. Both the progressive and total motility decreased over time during storage at 17 °C. Importantly, proline delayed the reduction of motility, although it did not prevent the decrease completely (see Appendix A). On day 5, addition of 20, 50, or 75 mM of proline to boar semen significantly improved both total motility and progressive motility (Figure 1A,B, *p* < 0.05). Among the treatments, 75 mM was the optimal one for maintaining sperm motility. Moreover, both the total motility and progressive motility in the group of 75 mM proline (TM—72.26 ± 0.74%; PM—59.04 ± 2.19%) were significantly higher than those in the control group (TM—60.67 ± 1.98%; PM—51.29 ± 1.47%) on day 9 of storage (see Appendix A). In addition, proline also increased the kinematic parameters, such as the straight-line velocity (VSL), average path velocity (VAP), and beat cross-frequency (BCF), on day 5 (see Appendix A).

#### 3.1.2. Proline Improves the Integrity of the Membrane and Acrosome

The status of the membrane and acrosome was evaluated based on the fluorescence staining (see Appendix A). As shown in Figure 1C, the percentage of sperm with intact plasma membrane in the 75 mM proline group was higher than that in the control group. The addition of 50, 75, and 100 mM proline improved acrosome integrity (Figure 1D, *p* < 0.05). The value of both the membrane and acrosome integrity was the largest in the group of 75 mM proline.

#### 3.1.3. Proline Increases the Mitochondrial Membrane Potential and ATP Levels

The level of MMP was measured through a dual-emission potential-sensitive probe JC-1. It is a green-fluorescent (λex 530 nm) monomer at a low membrane potential while a red-fluorescent (λem 590 nm) “J-aggregate” at a higher potential (Appendix A). The MMP level in the group of 75 mM proline was higher than that of the control (*p* < 0.05, Figure 1E). Similarly, the ATP level in the group of 75 mM proline was higher than that of the control. The addition of 75 mM proline led to a nearly 2-fold increase in ATP production relative to the control group (*p* < 0.05, Figure 1F). However, the addition of more proline did not yield higher values in MMP and ATP (Figure 1E,F).

### 3.2. Experiment II

#### Proline Modulates Redox Homeostasis

The status of the redox homeostasis in boar sperm was evaluated based on the fluorescence staining (see Appendix A). As shown in Figure 2A, the sperm ROS level decreased with the addition of proline. The ROS value in the group of 50, 75, and 100 mM proline was lower than that in control group (*p* < 0.05). Meanwhile, the addition of proline led to a decrease in the LPO level (Figure 2B). The addition of proline induced a nearly 2-fold increase in the activity of CAT as well as SOD at the concentration of 75 mM (Figure 2C,D). The group of 50 mM proline also promoted the activity of these two enzymes (*p* < 0.05). Moreover, supplementation with proline at 75 mM induced a significant promotion of the GSH level (Figure 2E, *p* < 0.05). Thus, the addition of proline modulated the redox homeostasis and protected sperm from ROS stress.

### 3.3. Experiment III

#### 3.3.1. Proline Improves the Motility, Integrity, and Redox Environment when Sperm are Exposed to Hydrogen Peroxide

To further uncover whether proline protects sperm against ROS attack, semen were exposed to hydrogen peroxide (H_2_O_2_). Exposure to 200 μM of H_2_O_2_ resulted in a significant reduction in motility and integrity, as well as aggravated oxidative damage to the membranes. Interestingly, proline rescued the reduction in total (Figure 3A) and progressive motility (Figure 3B). Meanwhile, the addition of proline maintained the integrity of the membrane (Figure 3C) and acrosome (Figure 3D), and MMP (Figure 3E) under oxidative stress. As expected, exposure to H_2_O_2_ led to a higher value in ROS and LPO levels, and lower value in GSH (Figure 3F–H, *p* < 0.05). Notably, the incubation of sperm with proline blocked the increase in the ROS and LPO levels, and maintained the GSH level (Figure 3H). These observations demonstrate that proline acts as an antioxidant by modulating the redox environment.

#### 3.3.2. Proline Improves the Motility, Integrity, and Redox Environment During Rapid Cooling

Boar sperm were exposed to rapid cooling, which resulted in a reduction in sperm quality in terms of motility, integrity of the acrosome and plasma membrane, and accumulation of ROS and LPO (Figure 4A–G). In this study, proline was supplemented to the extender to explore whether it has a protective effect on boar sperm during rapid cooling. The addition of 50 mM proline scavenged ROS accumulation (Figure 4F), and enhanced the GSH level (Figure 4G), which, in turn, reduced LPO (Figure 4H) and improved sperm quality (Figure 4A–G).

### 3.4. Experiment IV

To explore whether the secondary amine of pyrrolidine is the key factor for the protective role of proline against oxidative stress, the proline analogs that contain the secondary amine or not were added to the extender. Among those analogs, DL-pipecolinic acid (DL-PCA) was selected as a positive substrate because it contains a secondary amine structure while tetrahydro-2-furoic acid (THFA) as a negative one. Addition of DL-PCA rescued the reduction of progressive motility and total motility caused by H_2_O_2_, and THFA failed to provide a significant protection (Figure 5). These data demonstrate that the secondary amine structure is necessary for ROS scavenging.

### 3.5. Experiment V

#### Proline Protects Sperm against ROS via Proline Dehydrogenase

Western blot analysis showed that PRODH was expressed in boar sperm (Figure 6A). Moreover, the immunofluorescence assay indicated that PRODH was distributed in the acrosome region and the flagellum of the sperm tail (Figure 6B,C).

Proline at 75 mM increased the activity of PRODH (Figure 7A, *p* < 0.05). As shown in Figure 7B, the H_2_O_2_-induced reduction in progressive motility was attenuated by proline. However, the exposure of sperm to both proline and THFA decreased the protective role of proline in the presence of H_2_O_2_. In addition, in the absence of H_2_O_2_, the addition of THFA did not cause alteration in the progressive motility when compared to the group in the absence of THFA (that is, -THFA, -H_2_O_2_). These results indicated that inhibition of the activity of PRODH abolished the antioxidant function of proline.

## 4. Discussion

Accumulation of ROS occurs during liquid storage of boar semen, which results in oxidative stress to sperm when it overwhelms the antioxidant defense [9]. Hence, the addition of extrinsic antioxidants may efficiently protect sperm from oxidative stress and extend their lifespan during storage. In this study, we found that proline improved boar sperm motility, acrosome and membrane integrity, and decreased ROS levels during liquid storage. Here, for the first time, we identified that PRODH localized in the acrosome and the middle piece of the sperm tail, and found that the addition of proline to extender improved sperm quality by reducing ROS-induced damages.

In this study, we found that proline improved sperm motility during liquid storage, which is consistent with the previous studies in cooled-preserved donkey sperm [30] and in cryopreserved ram [31,32,33,34,35,36] and cynomolgus monkey [37] sperm. Moreover, proline also improved sperm membrane and acrosome integrity in this study, which is in agreement with earlier reports conducted in ram [31], cynomolgus monkey [37], and canine [39] sperm during cryopreservation. On the contrary, Ollero et al. [33] observed that proline did not improve the plasma membrane and acrosome integrity of ram sperm during cryopreservation (Fisher’s medium). The diverse results on the ram sperm membrane and acrosome integrity may be caused by the different extenders used in these studies.

In the present study, we observed that sperm ROS accumulated during storage and the fast cooling progress, which led to oxidative stress to sperm. The addition of proline to the extender improved sperm quality via attenuating the ROS stress, which is consistent with the report in ram sperm [31]. Moreover, the activity of antioxidant enzymes, such as CAT and SOD, were significantly increased by the addition of proline, which is in accordance with the previous findings in worms [43]. Zarse et al. [43] reported that catabolism of proline was proposed to generate transient ROS signals that activated homologues of p38 MAP kinase and Nrf2 (nuclear factor erythroid 2–related factor 2) in worms, leading to increasing expression of antioxidant enzymes.

To explore the mechanism, we exposed sperm to H_2_O_2_ and proline analogues at 37 °C. Interestingly, exposure to DL-pipecolinic acid (DL-PCA), which shared the secondary amine with proline, maintained the progressive motility, while THFA, which did not have the secondary amine, failed to do so. Moreover, Matysik et al. [27] reported that the secondary amine structure of pyrrolidine in proline, which had a low capability to provide an electron, was the key feature for quenching singlet oxygen in somatic cells. Therefore, the antioxidative property of proline in sperm relies on its secondary amine structure.

Proline dehydrogenase, a flavin adenine dinucleotide-containing enzyme, catalyzes the first step of proline catabolism, which forms D1-pyrroline-5-carboxylate (P5C) [44,45]. P5C is then converted to glutamate by Δ1-pyrroline-5-carboxylate dehydrogenase (P5CDH) or back to proline by cytosolic P5C reductase with either NADH or NADPH as cofactors [46,47]. In the present study, for the first time, we found that PRODH was localized in the flagellum of sperm, which was consistent with previous studies who found that PRODH bound tightly with the mitochondrial inner membrane in somatic cells [48,49]. Importantly, the activity of PRODH in sperm increased during storage with the addition of proline. Furthermore, inhibition of the activity of PRODH with THFA abolished the protective effect of proline under ROS stress. A previous study showed that PRODH helps cells to maintain ATP levels under nutrient deprivation; thus, proline metabolism appears to influence cellular ATP during oxidative and nutrient stress [50]. In this study, we also found an increase of ATP levels with the addition of proline after storage for up to 5 days. Thus, proline dehydrogenase was essential for the protective effect of proline on sperm. It would be interesting to uncover the underling mechanism by which proline enhances the antioxidative enzymes and thus prevents sperm from ROS stress in detail.

In addition, we found that PRODH was also distributed in the acrosome region. A recent study reported that mitochondria-derived membranes were assembled into the acrosome in the process of sperm generation, which provides a possible explanation for this distribution in the acrosome region [51]. The significance of PRODH in the acrosome needs to be elucidated.

## 5. Conclusions

Proline improved the quality of boar sperm during storage by modulating the redox environment. For the first time, we demonstrated that PRODH is localized in the acrosome and the middle piece of the sperm tail, and that proline protects sperm against ROS stress through PRODH-mediated metabolism and the secondary amine structure of pyrrolidine.

## Figures and Tables

**Figure 1 animals-10-01549-f001:**
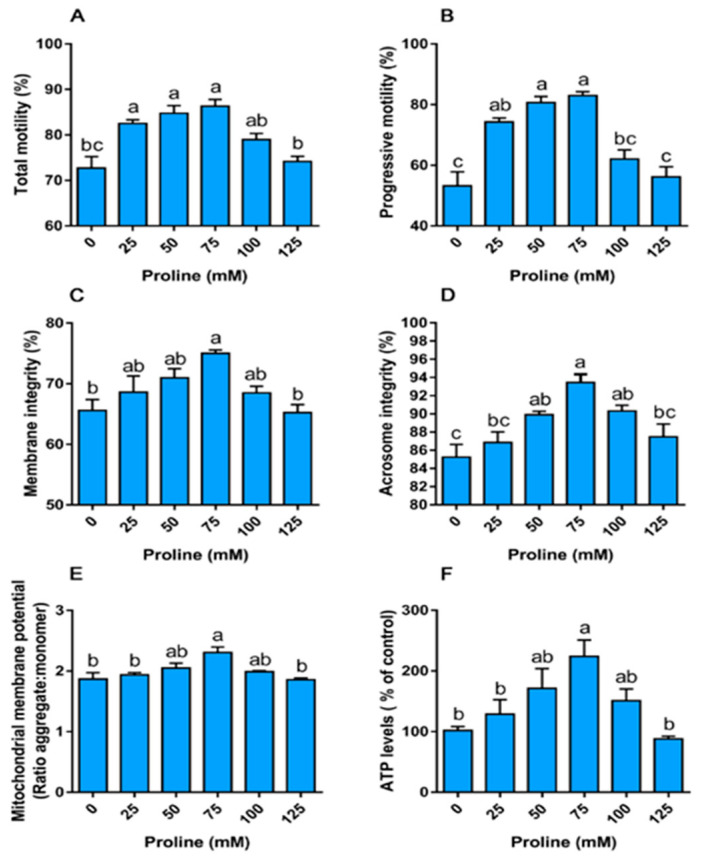
Effect of proline on the motility (**A**,**B**), integrity (**C**,**D**), and function (**E**,**F**) of boar sperm during liquid storage at 17 °C. Samples were supplemented with different concentrations of proline just after ejaculation and detected at day 5 of storage. (**A**) Total motility. (**B**) Progressive motility. (**C**) Membrane integrity. (**D**) Acrosome integrity. (**E**) Mitochondrial membrane potential, results are expressed as the ratio of JC-1 aggregate: monomer. (**F**) ATP levels. Values are shown as mean ± SEM of three independent experiments. Columns with nonidentical superscripts are significantly (*p* < 0.05) different from each other in statistics.

**Figure 2 animals-10-01549-f002:**
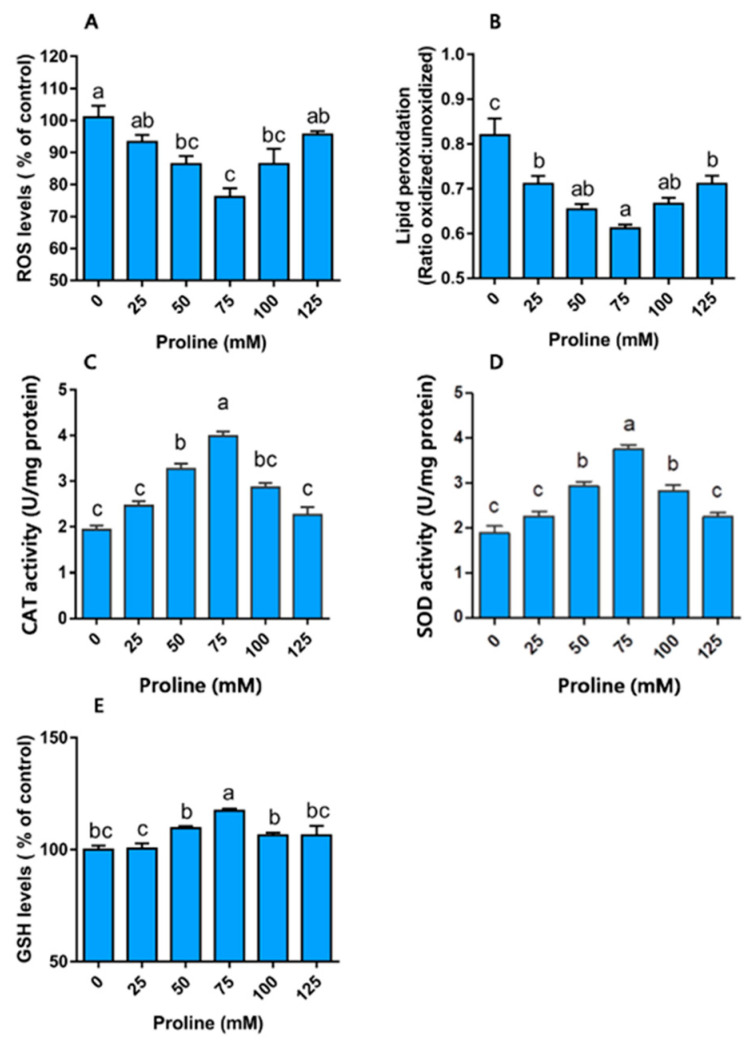
Effect of proline on the redox environment of boar sperm during storage at 17 °C. Samples were supplemented with different concentrations of proline just after ejaculation and detected on day 5 of storage. (**A**) ROS levels in sperm on day 5 of storage. (**B**) Lipid peroxidation levels, results are expressed as the ratio of oxidized forms: unoxidized forms. (**C**) GSH levels. (**D**) Activity of catalase. (**E**) Activity of superoxide dismutase. Values are shown as mean ± SEM of three independent experiments. Columns with nonidentical superscripts are significantly (*p* < 0.05) different from each other in statistics.

**Figure 3 animals-10-01549-f003:**
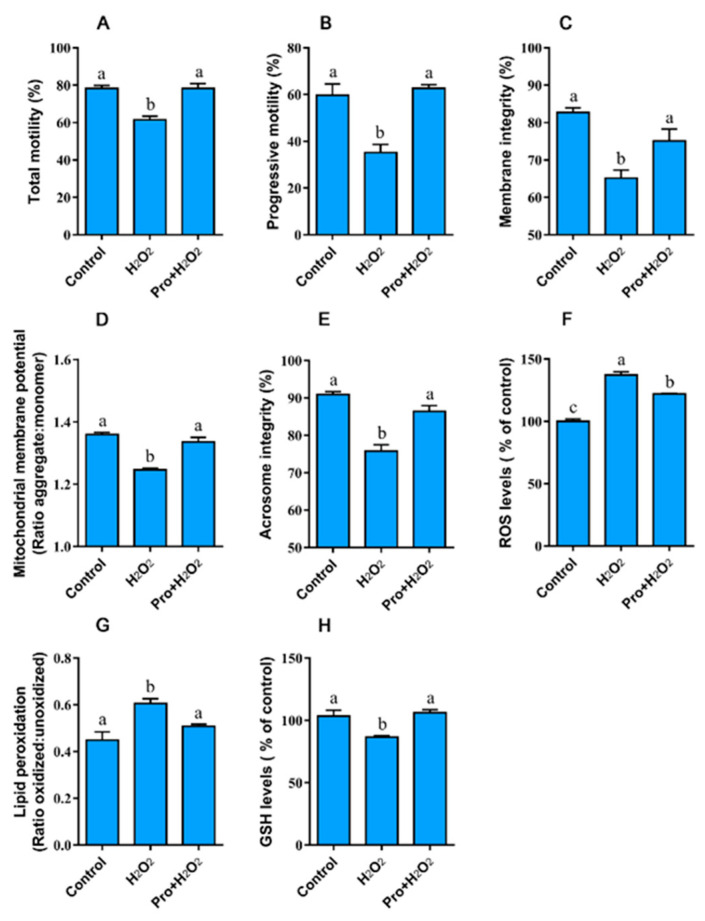
Effect of proline on the motility (**A**,**B**), integrity (**C**,**E**), function (**D**), and redox environment (**F**–**H**) of boar sperm under H_2_O_2_ stress. Samples in the control group were incubated with Modena at 37 °C for 2 h while others were incubated with 200 μM H_2_O_2_ in the absence and presence of 75 mM proline, respectively. (**A**) Total motility. (**B**) Progressive motility. (**C**) Membrane integrity. (**D**) Mitochondrial membrane potential, results are expressed as the ratio of JC-1 aggregate: monomer. (**E**) Acrosome integrity. (**F**) ROS levels. (**G**) Lipid peroxidation levels, results are expressed as the ratio of oxidized forms: unoxidized forms. (**H**) Glutathione levels. Values are shown as mean ± SEM of three independent experiments. Columns with nonidentical superscripts are significantly (*p* < 0.05) different from each other in statistics.

**Figure 4 animals-10-01549-f004:**
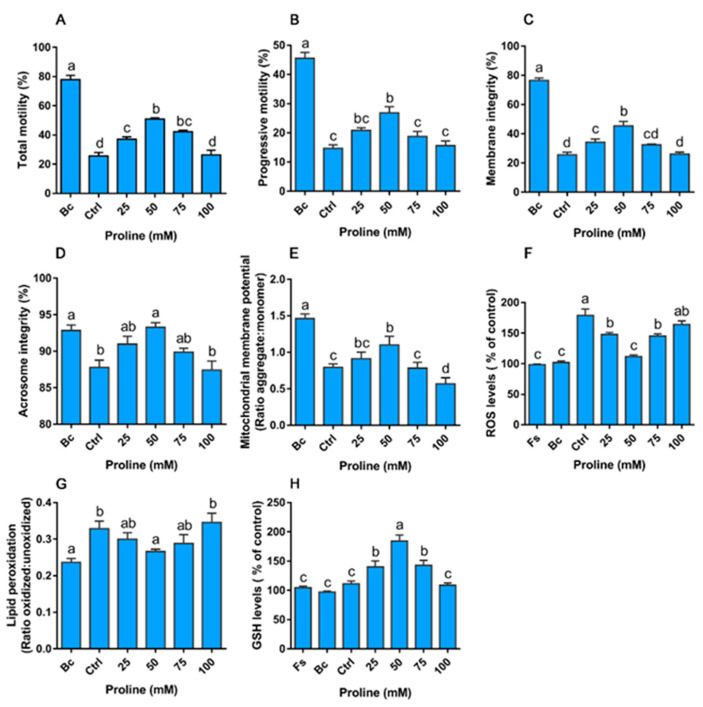
Effect of proline on the function (**A**,**B**), integrity (**C**–**E**), and redox environment (**F**–**H**) of boar sperm during rapid cooling. Samples were supplemented with different concentrations of proline before rapid cooling (from 30 to 5 °C) while samples in the control group were incubated at 30 °C all the time. After that, boar semen was sampled for the subsequent detection. (**A**) Total motility. (**B**) Progressive motility. (**C**) Membrane integrity. (**D**) Mitochondrial membrane potential, results are expressed as the ratio of JC-1 aggregate: monomer. (**E**) Acrosome integrity. (**F**) ROS levels. (**G**) Lipid peroxidation levels, results are expressed as the ratio of oxidized forms: unoxidized forms. (**H**) Glutathione levels. Values are shown as mean ± SEM of three independent experiments. Columns with nonidentical superscripts are significantly (*p* < 0.05) different from each other in statistics. Ctrl, control; Bc, before cooling; Fs, fresh semen.

**Figure 5 animals-10-01549-f005:**
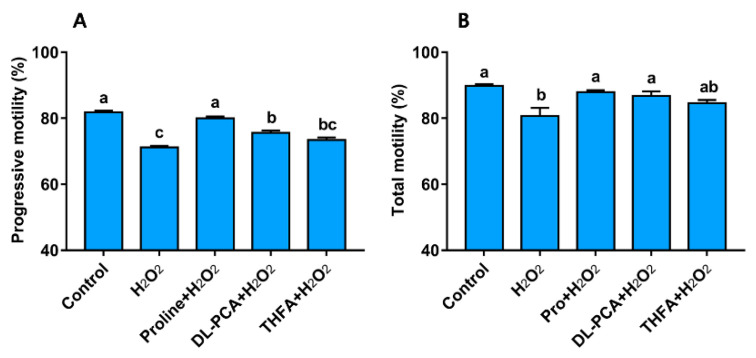
Effect of proline analogs on the progressive motility (**A**) and total motility (**B**) of boar sperm during H_2_O_2_ exposure. Except for semen in the control group, all samples were incubated with 200 μM H_2_O_2_ at 37 °C for 2 h in the presence of different proline analogs (5 mM). Values are shown as mean ± SEM of three independent experiments. Columns with nonidentical lowercase letters are significantly (*p* < 0.05) different from each other in statistics.

**Figure 6 animals-10-01549-f006:**
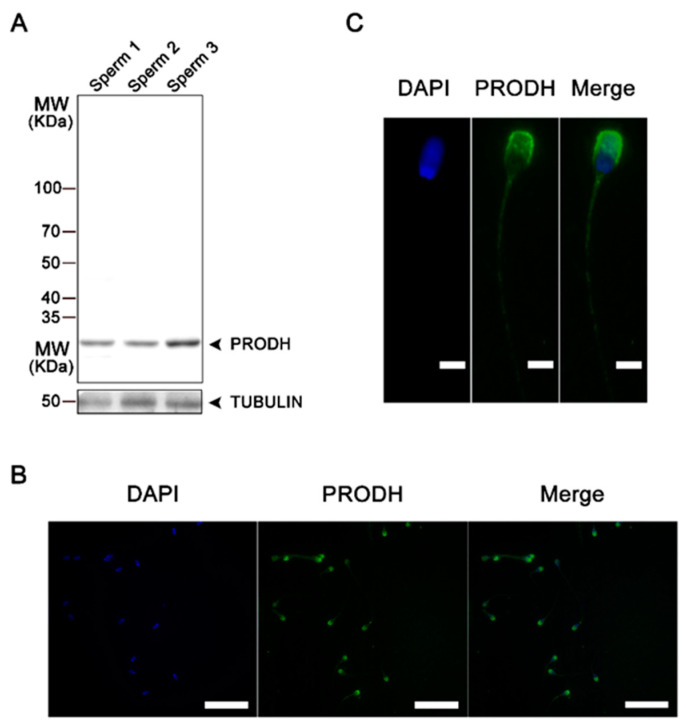
Expression of proline dehydrogenase (PRODH) in boar sperm. (**A**) Identification of PRODH in sperm by Western blot. Protein samples were obtained from three different boars. (**B**) Images of the immunofluorescent localization of PRODH in sperm at 400× magnification. Scale bars represent 100 µm. (**C**) Images of the immunofluorescent localization of PRODH in sperm at 1000× magnification. Scale bars represent 20 µm.

**Figure 7 animals-10-01549-f007:**
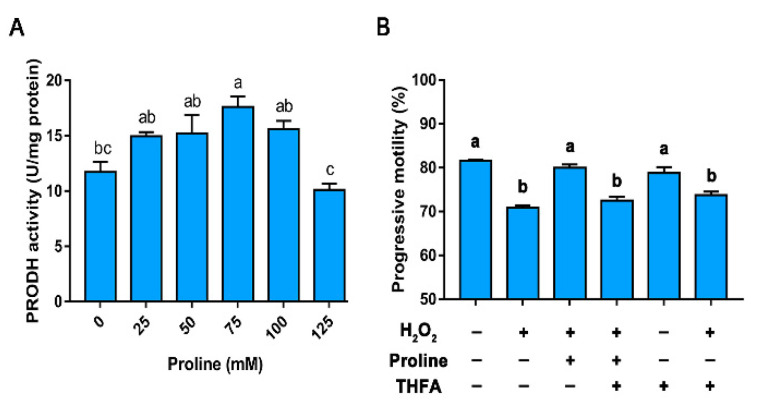
The role of proline dehydrogenase in the antioxidant properties of proline. (**A**) Effect of proline on the activity of PRODH on day 5 of storage. (**B**) Effect of inhibition of proline dehydrogenase on the progressive motility of sperm during H_2_O exposure. Samples were treated with or without H_2_O_2_ in the presence of proline or THFA and incubated at 37 °C for 2 h. Columns with nonidentical superscripts are significantly (*p* < 0.05) different from each other in statistics.

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
