# Peer review of "Proline Protects Boar Sperm against Oxidative Stress through Proline Dehydrogenase-Mediated Metabolism and the Amine Structure of Pyrrolidine"

_animals, 2020, doi:10.3390/ani10091549_

Round 1

Reviewer 1 Report

Specific comments, critiques and suggestions for improvement are below.

Line 70: Please use the word ‘evidence’ not ‘evidences’.

Section 2.3. Was the semen used for the experiments collected from the boars for three consecutive weeks (i.e. line 253 “All parameters were evaluated for at least three times with independent samples.”) In other words, were the three independent samples collected over three consecutive weeks?

Line 90: Please justify why the samples (n = 5) were mixed together to avoid individual differences. Although this reviewer acknowledges the increased workload of evaluating each individual boar, it would be important to analyze boars individually since there is the potential for one boar to possibly have an immense effect on the results.

Line 93: Use the word ‘following’ instead of ‘followed’.

Question: Why was the extended semen stored for 13 days? All the analyses seem to be performed at 5 days of storage.

Line 95: Please do not start a sentence with ‘And’.

Line 103: What was the volume of each fraction?

Line 112: Approximately how many total sperm cells were evaluated for motility per analysis?

Question: Why were the ROS levels, LPO levels, GSH levels and activity of
CAT and SOD evaluated only at the fifth day of storage after treatment? Should these levels also be evaluated on the 13th day of storage after treatment?

Section 2.5.1. What Reactive Oxygen Species were measured using this assay kit?
Section 2.9. The statistical analyses should include the model used and the components
of the model. In addition, it needs to be stated if repeated measures
analysis were used since the samples were evaluated over time.

Line 313: Should the word ‘elucidate’ be used? This reviewer is not familiar with the
word ‘elumidate’.

Line 332: Should the word ‘exposed’ be used instead of ‘susceptible’?

Figure 4: Please put in the figure 4 legend text the definitions of ‘Bc’ and ‘Fs’.

Lines 366/367: Insert the word ‘of’ between ‘exposure’ and ‘sperm’.

Line 367: Use the word ‘decreased’ instead of ‘vanished’.

Lines 398 to 400: This sentence is confusing to this review, please reword this sentence.
Perhaps this sentence can be separated into two sentences?

Line 420: Please insert a space between ‘effect’ and ‘of’.

Line 420: Insert the letter ‘A’ before ‘Previous’ at the beginning of the sentence.

Reviewer 2 Report

Proline seems to be a potent substance in improving many facets of sperm quality in general. The authors must be complemented by the wide range of technologies applied to establish if Proline addition is beneficial to protect sperm during short term storage at 4C.

However, most of the results relate to both the percentage sperm motility and the percentage progressive sperm motility and here the elaboration of CASA is insufficient. In order to establish that CASA results are valid the details need to be clearly stated. No mention is made of the the hardware aspects, temperature stage, control of variables and what is the frame rate, number of images sampled per sperm, how progressive sperm is calculated? Which type of cambers /slides have been used for motility measurements? Using CASA means you can analyze different sperm populations such as rapid progressive.

Furthermore it is highly surprising that a CASA system has been used but no mention has been made in terms of the kinematic parameters such as VCL, VSL, VAP etc. Leave alone the sub-population approach

Lines 230 - 240: Immunofluorecence of what?

Lines 268 - 271: Rephrase.. not clear what this means

Line 338: Fig. 4. May be I missed it but what Bc??

Line 427: Clarify...elaborate to make it succinctly clear

Round 2

Reviewer 2 Report

The authors have now made the most important changes. While I think that the paper is in order now for publication in Animals, I want to strongly suggest that in future the authors look at sub-populations of sperm as related to percentage rapid, medium and slow sperm. Looking at mere averages often defeats the purpose because of the very wide distribution of for example sperm swimming speeds.